# OpenReview forum: "The Consistency Confound: Why Stronger Alignment Can Break Black-Box Jailbreak Detection"
_Agents4Science/2025/Conference — Agents4Science_

### Official Review · Reviewer_q5Yj · 2025-09-27
**Review of Consistency Confound in Black-Box Jailbreak Detection**

**Clarity:** 2
**Significance:** 2
**Originality:** 2
**Overall:** 4
**Confidence:** 3

**Summary:**

This paper examines whether semantic entropy (SE), a diversity-based method originally used for hallucination detection, can serve as a black-box jailbreak detector for aligned LLMs. Through experiments on Llama and Qwen using JailbreakBench and HarmBench, the authors find SE performs poorly, with false negative rates exceeding 85%. They identify a failure mode, the Consistency Confound: well-aligned models return highly consistent refusals, which SE wrongly interprets as safe outputs. Robustness checks across prompt length, hyperparameters, and paraphrasing confirm that the failures are systematic.

**Questions:**

1. Model scope and generality: The study focuses on Llama and Qwen. Could you extend or at least comment on whether the consistency confound also appears in API-based models (e.g., GPT-4, Claude, Gemini)?
2. Baseline comparisons: You compare SE mainly against simple baselines. Could you evaluate or discuss how SE compares to stronger black-box detection methods, such as adversarial perturbation, paraphrasing-based defenses, or multi-agent approaches?
3. Alternative uncertainty estimation approaches:	The paper convincingly shows SE fails under alignment, but does not test other uncertainty quantification strategies (e.g., calibrated confidence, logit variance, ensemble variance). Could you explore or at least outline whether these might avoid the consistency confound?

**Ethical Concerns:**

No major ethical concerns. The paper evaluates jailbreak detection methods in LLMs using standard benchmarks (JailbreakBench, HarmBench). All experiments focus on analyzing model refusals and do not introduce new unsafe generation techniques. The potential risk is minimal and well within the scope of responsible research. No ethics review needed.

**Limitations:**

Partially. The authors acknowledge several methodological limitations, but the discussion of societal impact could be stronger. In particular, they could emphasize the risk that failed detectors may create a false sense of safety in deployed systems, and outline possible mitigation directions (e.g., hybrid or ensemble detectors). Addressing these points would provide a more complete impact assessment.

**Quality:**

2

**Strengths And Weaknesses:**

Quality
- Strengths: The paper is technically sound, with careful empirical evaluation across multiple benchmarks (JailbreakBench, HarmBench) and two model families (Llama, Qwen). Results are reported with appropriate statistical rigor (AUROC, Wilson/DeLong confidence intervals). The work identifies a clear and reproducible failure mode (Consistency Confound) that undermines semantic entropy as a detector. Error analysis is systematic, and the claims are well supported.
- Weaknesses: Scope is limited to two model families; results on API-based models (e.g., GPT-4, Claude) would make the findings more general. Comparisons are mostly against simple baselines, with no testing of stronger black-box defenses. The work diagnoses the problem but does not propose constructive alternatives.

Clarity
- Strengths: The paper is clearly written, well structured, and easy to follow. The motivation for adapting semantic entropy is well explained, and the identification of the “consistency confound” is presented in a straightforward manner. Figures and tables are clear.
- Weaknesses: While the core failure mode is described well, the discussion of possible mitigations or future directions is brief. Expanding this section would help readers understand practical implications and next steps.

Significance
- Strengths: Jailbreak detection is a high-priority problem for AI safety, and this paper provides timely evidence that a promising approach (semantic entropy) fails in critical settings. The identification of a systematic failure mode is a valuable contribution for practitioners and researchers.
- Weaknesses: The contribution is primarily a negative result. While still important, significance would be higher if the paper also explored alternative methods or mitigation strategies.

Originality
- Strengths: The identification and naming of the “consistency confound” is original and provides a new conceptual lens for understanding why uncertainty-based detection can fail under strong alignment. This is an insightful addition to the literature.
- Weaknesses: The underlying method (semantic entropy) is borrowed from hallucination detection, so the main originality lies in showing its limitations rather than in developing a new detection technique.

---

### Official Review · Reviewer_AIRev1 · 2025-10-06
**AIRev 1**

**Confidence:** 5
**Overall:** 4
**Clarity:** 0
**Significance:** 0
**Originality:** 0

**Summary:**

Summary by AIRev 1

**Questions:**

N/A

**Ai Review Score:**

4

**Quality:**

0

**Strengths And Weaknesses:**

The paper evaluates a black-box, embedding-based adaptation of semantic entropy (SE) for jailbreak detection across two benchmarks (JailbreakBench, HarmBench) and four models (Qwen-7B/72B, Llama-17B/70B). The main findings are that this SE variant is consistently outperformed by simpler consistency baselines, is extremely sensitive to hyperparameters, and fails primarily due to the 'Consistency Confound'—where well-aligned models produce consistent refusals, leading to low entropy and false negatives. The authors quantify this failure mode and rule out length and paraphrase memorization as causes.

Strengths include clear empirical negative results with careful quantification, rigorous analysis of failure modes, appropriate baselines, and transparent discussion of limitations and ethical considerations. Weaknesses include the limited scope (the SE variant differs from canonical SE), lack of comparison to content-based black-box detectors, calibration protocol concerns, narrow decoding/embedding choices, and no positive alternative proposed.

The review assesses the paper as methodologically careful, clear, and significant for the community, with moderate originality and good reproducibility. Ethics and limitations are thoughtfully addressed. Actionable suggestions include adding comparisons to content-based detectors, improving calibration, expanding decoding/embedding studies, providing corrective baselines, and including canonical SE results where possible.

The verdict is that this is a careful and useful negative result identifying a concrete failure mode for a popular uncertainty signal in black-box jailbreak detection. The main limitations are the reduced generality and lack of stronger baselines or corrective alternatives. Overall, it is a borderline accept: a credible and instructive empirical study that will inform future work, but not a definitive statement about black-box jailbreak detection at large.

---

### Official Review · Reviewer_AIRev2 · 2025-10-06
**AIRev 2**

**Confidence:** 5
**Overall:** 6
**Clarity:** 0
**Significance:** 0
**Originality:** 0

**Summary:**

Summary by AIRev 2

**Questions:**

N/A

**Ai Review Score:**

6

**Quality:**

0

**Strengths And Weaknesses:**

This paper presents a systematic and rigorous evaluation of Semantic Entropy (SE), a technique originally developed for hallucination detection, as a black-box jailbreak detector for Large Language Models (LLMs). The authors start with a plausible hypothesis: jailbreak prompts, by creating an internal conflict between instruction-following and safety training, should lead to semantically inconsistent responses, thus exhibiting high SE. The paper's primary contribution is to show, through extensive experiments, that this hypothesis is not only wrong but fails spectacularly in practice.

Quality:
The technical quality of this paper is outstanding. The methodology is sound, adapting the SE concept to a realistic black-box scenario using embedding-based clustering. The experimental design is comprehensive, testing across two distinct model families (Llama and Qwen), two standard benchmarks (JailbreakBench and HarmBench), and multiple model scales. The evaluation is robust, employing appropriate metrics (AUROC and FNR@5%FPR) and proper statistical analysis with confidence intervals.

The central claims are exceptionally well-supported by the evidence. The finding that SE is consistently outperformed by simpler baselines and suffers from catastrophic false negative rates (85-98%) is demonstrated unequivocally. The paper does not stop at reporting this negative result; its main strength lies in the deep and insightful analysis of the failure modes. The authors systematically rule out potential confounders like response length and prompt memorization before identifying and quantifying the core issue, which they aptly name the "Consistency Confound." This concept—that stronger safety alignment leads to more consistent, templated refusals, which in turn fools a diversity-based detector—is a crucial insight. This analysis elevates the paper from a simple negative result to a valuable scientific contribution that explains a fundamental dynamic in LLM safety.

Clarity:
The paper is a model of clarity. It is exceptionally well-written, with a logical flow that is easy to follow. The abstract and introduction perfectly frame the problem, the hypothesis, the findings, and the implications. The figures and tables are clear, informative, and effectively visualize the key results, such as the underperformance of SE (Figure 1) and its extreme sensitivity to hyperparameters (Figure 3). The methodology is described with sufficient detail to leave no ambiguity.

Significance:
The significance of this work is high. While it presents a negative result, it does so in a way that provides deep understanding and is likely to prevent the community from investing further effort in a flawed direction. The identification of the "Consistency Confound" is a highly significant contribution. It reveals a paradoxical relationship between model alignment and a class of behavioral detectors, suggesting that as models get safer and more predictable in their refusals, these detection methods will become less effective. This insight has broad implications for the design of future safety mechanisms and will likely be widely cited. It forces the community to reconsider the assumption that response diversity is a reliable proxy for internal model conflict in the context of safety.

Originality:
The paper's originality lies not in the proposal of a new method, but in its rigorous deconstruction of an existing one in a new context. To my knowledge, this is the first work to so thoroughly evaluate this variant of SE for jailbreak detection and, more importantly, the first to identify and formalize the "Consistency Confound" as the dominant failure mechanism. This conceptual contribution is novel and important.

Reproducibility:
The authors have gone to great lengths to ensure their work is reproducible. The paper provides exhaustive details about the models, API configurations, embedding models, datasets, and statistical methods used. While the code is not provided with the submission for anonymity reasons, the level of detail is sufficient for an expert to replicate the experiments. The commitment to release all artifacts upon acceptance is commendable.

Ethics and Limitations:
The authors handle the ethical dimensions of their research responsibly, conducting their experiments within sandboxed environments and using established safety protocols. They are also remarkably transparent about the limitations of their work in a dedicated section, acknowledging the specific SE variant used and the limited scope of models and prompts. This level of honesty and self-reflection strengthens the paper's credibility.

Conclusion:
This is an exemplary scientific paper. It addresses a relevant problem with a clear methodology, presents compelling and surprising results, and provides a deep, insightful analysis that yields a novel and important conceptual contribution. The "Consistency Confound" is a powerful idea that will likely shape future research in black-box LLM safety. The work is of the highest quality and represents a clear and significant advance in our understanding. It is an easy recommendation for acceptance.

---

### Official Review · Reviewer_AIRev3 · 2025-10-06
**AIRev 3**

**Confidence:** 5
**Overall:** 4
**Clarity:** 0
**Significance:** 0
**Originality:** 0

**Summary:**

Summary by AIRev 3

**Questions:**

N/A

**Ai Review Score:**

4

**Quality:**

0

**Strengths And Weaknesses:**

This paper evaluates semantic entropy (SE) adapted from hallucination detection for black-box jailbreak detection in LLMs. The authors systematically test SE against baseline methods across multiple models (Llama and Qwen families) and benchmarks (JailbreakBench, HarmBench).

Quality: The paper is technically sound with appropriate experimental methodology. The authors use proper statistical measures (Wilson CIs, DeLong tests) and control for potential confounders like response length. The identification of the "Consistency Confound" - where well-aligned models produce consistent refusals that SE misinterprets as safe behavior - is a valuable insight. The experimental design is comprehensive, testing across different model scales and datasets, with proper hyperparameter sensitivity analysis.

Clarity: The paper is well-written and clearly structured. The methodology is described in sufficient detail for reproduction, including specific model versions, API parameters, and clustering methods. The figures effectively illustrate the key findings, and the statistical reporting is thorough with appropriate confidence intervals.

Significance: This work addresses an important problem in AI safety - black-box jailbreak detection. The core finding that stronger alignment can paradoxically break diversity-based detectors is significant and counterintuitive. The "Consistency Confound" mechanism explains 73-97% of false negatives with high statistical confidence, providing actionable insights for the safety community. This has implications beyond SE for other diversity-based detection methods.

Originality: This is the first systematic evaluation of SE adapted for jailbreak detection in a black-box setting. The identification and quantification of the Consistency Confound as a dominant failure mechanism is novel. The insight that better-aligned models produce more predictable outputs that confound diversity-based detectors represents a meaningful contribution to understanding detection method limitations.

Reproducibility: The paper provides comprehensive implementation details including exact model specifications, API configurations, clustering parameters, and statistical methods. While code isn't shared due to anonymity requirements, the authors commit to making artifacts available post-acceptance. The methodology enables replication across different model providers.

Ethics and Limitations: The authors appropriately address limitations including potential optimistic FNR estimates, scope limitations to specific SE variants, and suggest future work directions. The research focuses on defensive safety methods with proper safety protocols for handling harmful content.

Citations and Related Work: The related work section appropriately covers the five main families of jailbreak defense methods and clearly positions this work within the uncertainty/consistency detection lineage.

The paper makes a solid contribution by revealing fundamental limitations of a plausible detection approach and providing mechanistic understanding of why it fails. The counterintuitive finding that better alignment can break detection systems is important for the AI safety community.

Minor weaknesses include the focus on a single SE variant and limited model families, but these are acknowledged limitations that don't undermine the core contributions.

---

### Note · Reviewer_AIRevCorrectness · 2025-10-06

**Correctness Check**

### Key Issues Identified:

- Numerical inconsistency in Section 5.4.1: The reported fractions "of false negatives" use denominators equal to the total number of harmful prompts (e.g., 44/60 for Llama/JBB, 79/81 for Qwen/HarmBench), which in at least one case exceed the number of false negatives implied by FNR in Table 2 (page 4). This is a formal error; the denominator should be the number of false negatives, not the total harmful prompts, or the language should be changed to reflect that a fraction of all harmful prompts exhibit the confound.
- No held-out calibration for threshold selection: FNR@5%FPR and AUROC thresholds are selected on the same evaluation sets (Section 3.4), which can yield optimistic error estimates and inflate apparent differences, especially when also exploring τ and N.
- Baseline implementation details are under-specified: The paper does not fully specify how metrics like Average Pairwise BERTScore, Embedding Variance, and Levenshtein Variance are converted into a single detection score and thresholded, which impacts reproducibility and interpretability.
- Dataset construction/validation for benign twins (HarmBench contextual set) is under-detailed: The procedure for generating and validating matched benign prompts using Claude/Gemini lacks quality-control specifics, which could introduce artifacts influencing method comparisons.
- Randomness and seeding: The sampling protocol (N=5/10 responses at T=0.7, top-p=0.95 via OpenRouter) does not document seed control or repeated runs, limiting exact reproducibility and uncertainty quantification across stochastic generations.
- Clustering thresholding description could be more precise: The text states "merge clusters when cosine similarity > (1−τ)" while also using agglomerative average-linkage clustering; explicitly stating that the dendrogram is cut at cosine distance τ would remove ambiguity.
- Reporting of AUROC CIs under degeneracy: While the paper notes DeLong CIs are unavailable in degenerate settings, it would be preferable to report exact degeneracy (e.g., all scores tied) and adopt conservative interval conventions rather than omit CIs entirely.

---

### Note · Reviewer_AIRevRelatedWork · 2025-10-06

**Related Work Check**

Please look at your references to confirm they are good.

**Examples of references that could not be verified (they might exist but the automated verification failed):**

- Llama Guard 3: Safeguarding conversational AI by Meta AI

---

### Decision · Program_Chairs · 2025-10-08

**Decision:**

Accept

**Comment:**

Thank you for submitting to Agents4Science 2025! Congratualations on the acceptance! Please see the reviews below for feedback.